# Transcriptomic Analysis Reveals Intrinsic Abnormalities in Endometrial Polyps

**DOI:** 10.3390/ijms25052557

**Published:** 2024-02-22

**Authors:** Christine Shan-Chi Chiu, Ling-Yu Yeh, Szu-Hua Pan, Sheng-Hsiang Li

**Affiliations:** 1Graduate Institute of Medical Genomics and Proteomics, College of Medicine, National Taiwan University, Taipei 100, Taiwan; shanchichiu@gmail.com; 2Department of Obstetrics and Gynecology, MacKay Memorial Hospital, Taipei 104, Taiwan; 3Department of Medical Research, MacKay Memorial Hospital, Tamsui District, New Taipei 251, Taiwan; lindyyeh2008@gmail.com; 4MacKay Junior College of Medicine, Nursing, and Management, Beitou District, Taipei 112, Taiwan; 5Department of Chemical Engineering and Biotechnology, National Taipei University of Technology, Taipei 106, Taiwan

**Keywords:** endometrial polyps, gene expression, Wnt signaling pathway, vascular smooth muscle, female infertility

## Abstract

Endometrial polyps (EPs) are benign overgrowths of the endometrial tissue lining the uterus, often causing abnormal bleeding or infertility. This study analyzed gene expression differences between EPs and adjacent endometrial tissue to elucidate intrinsic abnormalities promoting pathological overgrowth. RNA sequencing of 12 pairs of EPs and the surrounding endometrial tissue from infertile women revealed 322 differentially expressed genes. Protein–protein interaction network analysis revealed significant alterations in specific signaling pathways, notably Wnt signaling and vascular smooth muscle regulation, suggesting these pathways play critical roles in the pathophysiology of EPs. Wnt-related genes *DKK1* and *DKKL1* were upregulated, while *GPC3*, *GREM1*, *RSPO3*, *SFRP5*, and *WNT10B* were downregulated. Relevant genes for vascular smooth muscle contraction were nearly all downregulated in EPs, including *ACTA2*, *ACTG2*, *KCNMB1*, *KCNMB2*, *MYL9*, *PPP1R12B*, and *TAGLN*. Overall, the results indicate fundamental gene expression changes promote EP formation through unrestrained growth signaling and vascular defects. The intrinsic signaling abnormalities likely contribute to clinical symptoms of abnormal uterine bleeding and infertility common in EP patients. This analysis provides molecular insights into abnormal endometrial overgrowth to guide improved diagnostic and therapeutic approaches for this troublesome women’s health condition. Confirmation of expanded cohorts and further investigations into implicated regulatory relationships are warranted.

## 1. Introduction

Endometrial polyps (EPs) are a common condition among women of a reproductive age, with a prevalence ranging from 7.8% to 34.9% [1]. They are caused by the excessive growth of the glandular, stromal, and vascular components in the endometrial tissues. The stroma of EPs consists of large, thick-walled blood vessels and spindle-shaped cells with a fibroblast-like appearance [2]. Clinical symptoms include abnormal uterine bleeding and infertility [3,4]. Infertility may be caused by EPs forming a mechanical barrier that prevents embryo implantation or releasing biochemical factors unfavorable for implantation. A previous study has shown that EP patients exhibit elevated glycodelin levels in the uterine cavity, which may inhibit sperm capacitation and endometrial receptivity [5]. Notably, up to 35% of infertile women suffer from Eps [4].

The primary clinical treatment for EPs is hysteroscopic polypectomy with or without hormonal therapy [3,6]. A randomized controlled trial of infertile women undergoing a hysteroscopic polypectomy prior to intrauterine insemination showed that polyp removal improved pregnancy outcomes compared to no intervention [7]. Moreover, clinical pregnancy rates were significantly higher in patients who underwent hysteroscopy, polypectomy, and adhesion treatment compared to those who did not receive a hysteroscopy [8].

The pathogenic mechanisms underlying EPs remain partially elusive. Factors that can lead to EP formation include gene mutations, overexpression of endometrial aromatase, and aberrant proliferation of monoclonal endometrial cells [2]. As proposed by Indraccolo et al., EP development involves multiple pathogenic roles, including imbalanced estrogen treatment, overexpression of B-cell lymphoma-2 (BCL2), senescence, tamoxifen use, and obesity [9].

Regarding gene expression characteristics in polyp tissues, some studies showed that postmenopausal bleeding EP patients exhibited higher estrogen receptor, progesterone receptor, KI67, and BCL2 levels in the polyp epithelium and stroma compared to the adjacent endometrium [10]. Matrix metalloproteinase and glycodelin levels were also elevated in polyp tissues compared to the normal endometrium [11,12]. In proliferative EP samples, BCL2 expression was significantly increased, whereas progesterone receptor expression was weaker or absent [10]. Another study indicated that the co-expression of the progesterone receptor and G protein-coupled receptor 30 could be an essential mechanism in EP pathogenesis [13].

No comprehensive gene expression study currently compares endometrial polyps and adjacent tissues. Further elucidating the molecular mechanisms underlying EP growth may provide better clinical guidance for polyp and infertility treatments. Therefore, in this study, we used RNA sequencing (RNA-seq) to conduct an in-depth analysis of the gene expression differences between Eps and the adjacent endometrium in endometrial tissues at the mid-to-late proliferative phase of infertile women.

## 2. Results

### 2.1. Physiological Parameters of the Subject

This study included 12 cases who presented to the Tamsui infertility clinic of the Mackay Memorial Hospital and had EPs revealed by ultrasound. Surgery to remove the EPs was then scheduled. Study participants’ ages ranged from 36 to 43 years, had menstrual cycles lasting from 25 to 60 days, and had a BMI ranging from 19 to 24. AMH values ranged from 0.23 and 12.19. Polyps were surgically removed between days 7 to 13 of the menstrual cycle (Appendix A).

### 2.2. Transcriptome Analysis of Polyps and Adjacent Endometrium

To access the variations in gene expression between the tissues of polyps and the corresponding neighboring endometrium, we ran an RNA-seq analysis. The findings showed that 58,825 genes were detected in all specimens. Principal component analysis (PCA) showed that, regardless of whether they were polyp or adjacent endometrial tissue specimens, the overall gene expression patterns were relatively dispersed, with no significant differences observed between the two groups of tissues (Figure 1a). The volcano plot showed all 58,825 detected genes. Although the expression levels of most genes were quite similar, there were still 322 differentially expressed genes (DEGs) between the polyp group and the adjacent endometrial tissue group. The expression levels of these genes changed by over 1.5 fold (*p*-value < 0.05) between the two groups of specimens. Among them, 88 genes were significantly more expressed in the polyp group. In comparison, the other 234 genes had significantly higher expression levels in the adjacent endometrium (Figure 1b and Appendix A). The heatmap shows the differences between the two DEG groups, with distinct transcriptome profiles between the polyps and control groups (Figure 1c). Next, we selected 10 highly expressed genes, including *ACTG2*, *CNN1*, *DES*, *GPC3*, *HOXA13*, *LEFTY2*, *LMOD1*, *PTGIS*, *RERG*, and *TAGLN*, in the endometrium for quantitative reverse transcription-polymerase chain reaction (qRT-PCR) confirmation. The results are consistent with the trend of the RNA-seq data (Figure 1d).

### 2.3. Protein–Protein Interaction (PPI) Network Construction

The human PPI network in the STRING database was used to examine the interactions between DEG-encoded proteins. STRING analysis of all DEGs revealed that the PPI network contained 172 protein nodes and 160 edges (Figure 2). DEGs that do not appear in the nodes can be predicted genes, long non-coding RNAs that do not encode proteins, or genes not included in the STRING database. On the STRING platform, we utilized the k-means clustering algorithm to divide the 172 protein nodes into three clusters, each represented by a distinct color (red, green, blue), with 48, 73, and 51 nodes, and 28, 57, and 52 edges, respectively.

### 2.4. Cytoscape Analysis and the Kyoto Encyclopedia of Genes and Genomes (KEGG) Pathway Analysis

The correlations of the protein networks of the three subclusters were redrawn with the top 10 hub genes’ nodes and edges for each subcluster using Cytoscape software (version 3.10.1) based on STRING analysis results (Figure 3). The circle size represents the importance of that protein in the network (combined score). The color represents the fold change (FC = polyp tissue/control) of the RNA-seq results between the polyps and adjacent endometrium, with red indicating positive values and blue indicating negative values. Color depth was based on the magnitude of fold change differences. CytoHubba analysis identified 5 key hub genes in each subcluster, including *DKK1*, *GPC3*, *BMP7*, *FGF17*, *WNT10B* (Figure 3a), *CNN1*, *ACTG2*, *ACTA2*, *MYL19*, *TAGLN* (Figure 3c), and *NCAM1*, *SNAP25*, *CD36*, *KIRZDL1*, *SORBS1* (Figure 3e).

KEGG results show that essential pathways in each subcluster are the Wnt signaling pathway, cytokine–cytokine receptor interaction, hippo signaling pathway, axon guidance, TGF-beta signaling pathway, and glycosaminoglycan biosynthesis–heparan sulfate/heparin (Figure 3b); dilated cardiomyopathy, vascular smooth muscle contraction, hypertrophic cardiomyopathy, motor proteins, regulation of actin cytoskeleton, focal adhesion, cGMP-PKG signaling pathway, arrhythmogenic right ventricular cardiomyopathy, adrenergic signaling in cardiomyocytes, and cardiac muscle contraction (Figure 3d); and oxytocin signaling pathway and neuroactive ligand receptor interaction (Figure 3f). Among them, the Wnt signaling and dilated cardiomyopathy signaling pathways were the most significant.

GO (Gene Ontology) enrichment analysis for the biological process of the three PPI subclusters showed that DEGs in the first subcluster (Figure 3a) were mainly enriched in the regulation of the Wnt signaling pathway, including the canonical Wnt signaling pathway, positive regulation of the Wnt signaling pathway/planar cell polarity pathway, negative regulation of the canonical Wnt signaling pathway (Table 1), and positive regulation of the canonical Wnt signaling pathway (Appendix A). In addition, SMAD proteins downstream of the Wnt signaling pathway also appeared in the related enrichments. DEGs in the second subcluster (Figure 3c) showed significant enrichment in muscle contraction pathways, such as myofibril assembly and regulation of muscle contraction (Table 1). In addition, biological processes, including actin filament-based movement, actin filament organization, and actomyosin structure organization, were also found, although the *p*-values were not reached significantly (Appendix A). Moreover, the GO terms of the cellular composition and the molecular function also showed some muscle-related events, including Z disc, stress fiber, dynactin complex, actin filament, actin cytoskeleton, actin binding, and muscle alpha-actinin binding (Appendix A). The related genes in this subcluster were almost all downregulated in polyp tissues. The DEGs in the third subcluster (Figure 3e) mainly regulate ion transport and signal transduction across the plasma membrane. It appears challenging to integrate correlations from the biological processes involved (Table 1 and Appendix A).

Considering the pathways significantly enriched in polyp specimens, we further analyzed the specific genes involved in these pathways. The genes regulating the Wnt signaling pathway included *DKK1*, *DKKL1*, *WNT10B*, *GREM1*, *RSPO3*, *SFRP5*, and *GPC3*. *DKK1* and *DKKL1* showed a significant upregulation in polyp specimens, while *WNT10B*, *GREM1*, *RSPO3*, *SFRP5*, and *GPC3* showed a downward trend (Figure 4a and Appendix A). The genes involved in muscle differentiation or contraction pathways included *ACTC1*, *PLN*, *DES*, *ITGA10*, *ITGA8*, *SLC8A2*, *ACTA2*, *KCNMB1*, *KCNMB2*, *PPP1R12B*, *MYL9*, *ACTG2*, *TUBA3E*, *BUB1B-PAK6*, *COX6A2. BUB1B-PAK6* and *ITGA8* in muscle-related signaling pathways showed increased expressions in polyp tissues, while *COX6A2*, *TUBA3E*, *KCNMB2*, *ITGA10*, *ACTC1*, *MYL19*, *ACTA2*, *PPP1R12B*, *DES*, *PLN*, *SLC8A2*, *KVNMB1*, and *ACTG2* showed decreased expressions in polyp tissues (Figure 4b and Appendix A).

## 3. Discussion

Gene expression variations between polyp tissues and the surrounding endometrium were revealed using RNA-seq data. The abnormal overgrowth of polyps may be caused by changes in gene expression, even when their histological appearance is not significantly different from that of the endometrium. We examined the PPI networks of DEGs and found three polyp-associated protein groupings. The WNT signaling pathway and many pathways linked to muscle function were the most influential among them.

During the menstrual cycle, vascular smooth muscle cells (VSMCs) of the endometrium steadily proliferate to promote the angiogenesis of endometrial arterioles, enabling them to adjust arteriolar diameter to control blood flow rapidly [14]. Abnormal arteriogenesis may lead to menorrhagia and breakthrough bleeding.

Our findings are supported by the KEGG pathway analysis of the PPI network for subcluster proteins, which revealed significant differences in the vascular smooth muscle contraction pathway. Relevant critical genes were downregulated in endometrial polyps, including *ACTA2*, *KCNMB1*, *KCNMB2*, *PPP1R12B*, *MYL9*, and *ACTG2*. The regulation of muscle contraction, myofibril assembly, actin filament-based movement, actin filament organization, actomyosin structure organization, Z disc, stress fiber, dynactin complex, actin filament, actin cytoskeleton, and actin-binding were among the other muscle function-related signaling pathways that were found to be significantly impacted by the GO enrichment analysis.

Specific markers, such as smooth muscle alpha-actin (α-SMA), calponin-1 (CNN1), caldesmon (CALD1), tropomyosin alpha-1 (TPM1), γ-smooth muscle actin (ACTG2), and myosin heavy chain 1 (MYH1) are expressed during the differentiation of VSMCs [15,16]. In the polyp specimens, the expression of these indicators’ genes was downregulated. Additionally, there was a declining tendency in the polyp specimens for factors that were important in arteriogenesis and arterial remodeling, such as transforming growth factor beta-1-induced transcript 1 (*TGFB1I1*), angiopoietin-1 (*ANGPT1*), and vascular endothelial growth factor A (*VEGFA*) [17].

Microvessels acquire a covering layer of VSMCs during arteriogenesis, enabling rapid blood flow regulation through internal radius changes [14]. VSMCs can differentiate from the surrounding matrix, epithelial, and endothelial cells [18,19]. We also observed changes in cell migration-related pathways in the GO enrichment analysis, including mesenchyme migration, integrin-mediated signaling pathway, focal adhesion, integral component of the membrane, and integrin complex.

Transgelin (TAGLN) is an actin cross-linking/gelling protein widely expressed in smooth muscles and fibroblasts, and it is susceptible to transformation and shape changes. Because it promotes actin polymerization, TAGLN is thought to be an early marker of smooth muscle differentiation in vascular and visceral smooth muscles. We observed a significant decrease in TAGLN gene expression in polyp tissues, which may indicate a decrease in actin polymerization and smooth muscle development. Actin polymerization is also a prerequisite in embryo implantation, and its absence can lead to defective embryo implantation [20]. Furthermore, while TGF-β controls TAGLN expression, the dysregulation of the TGF-β pathway in polyp tissues (see Figure 3b) can be a factor in TAGLN downregulation.

One of the most prevalent primary complaints for individuals with polyps in their endometrium is infertility or abnormal uterine bleeding [21]. As mentioned above, we deduced that aberrant bleeding could be facilitated by the downregulation of relevant genes in polyp tissues, which could prevent smooth muscle cell development and arteriogenesis. Infertility can also arise from these structural defects, as they can impact the implantation and development of the embryo following fertilization. According to a prior study, endometrial spiral arterioles are crucial for regulating the menstrual cycle and the subsequent implantation and placentation of embryos [16].

The Wnt signaling pathway, which regulates cell proliferation and migration in healthy tissues, embryos, and cancer cells, is a highly conserved intracellular signaling cascade [22,23]. According to a previous study, endometrial polyp tissues have an increased expression of WNT1. Progesterone was confirmed to inhibit endothelial cell activity, induce apoptosis via the Wnt pathway, and thus ameliorate the endometrial polyp [2]. Our results demonstrate that the Wnt pathway is substantially altered in polyp specimens. Significant dysregulation of the expression of critical genes, including *DKK1*, *GPC3*, *BMP7*, *FGF17*, and *WNT10B*, was observed in the PPI network; this dysregulation could potentially be associated with the proliferation of polyp tissues.

WNT proteins belong to a large family of secreted signaling molecules that signal by binding to a receptor complex. To date, at least three Wnt signaling pathways have been described. The activation of the canonical Wnt pathway disrupts complex binding and releases β-catenin, which then translocates to the nucleus and activates target gene expression [24]. Non-canonical pathways include Wnt/planar cell polarity and Wnt/calcium signaling. The activation of the Wnt/calcium pathway leads to elevated intracellular Ca^2+^ levels, activating calcium-sensitive phosphatase calcineurin, which dephosphorylates and promotes the nuclear import of the transcription factor nuclear factor of activated T cells [25].

The Wnt pathway can participate in regulating cardiac and vascular development during embryogenesis. In the canonical Wnt pathway, downstream β-catenin signaling affects endothelial cell differentiation, thereby altering vascular remodeling and arteriovenous specification [26]. Additionally, studies have found that noncanonical Wnt signaling can regulate endothelial cell proliferation and vascular network formation [27,28]. Several reports also described the paracrine effects of Wnt ligands in angiogenesis [29,30].

The Wnt pathway encompasses many proteins and synergistically regulates with other pathways. In transformed mammary epithelial cells, TGF-β and Wnt signaling cooperatively induce the initiation of the epithelial–mesenchymal transition [31]. The expression of these complex proteins can affect Wnt signal transduction. Secreted frizzled-related protein 5 (SFRP5) is an identified anti-inflammatory adipokine belonging to the SFRP family [32]. SFRP proteins antagonize the Wnt pathway through competitive inhibition [33], typically by directly binding Wnts and blocking both canonical and noncanonical Wnt signaling [34]. In our polyp specimens, the expression of this gene was significantly decreased, which could be related to Wnt pathway abnormalities leading to excessive growth.

Glypican-3 (GPC3) is a member of the heparan sulfate proteoglycans family that attaches to the extracellular side of the cell membrane via a glycosylphosphatidylinositol anchor. GPC3-expressing cells exhibit an epithelial morphology with an altered cytoskeletal structure, decreased migratory and clonogenic abilities, increased homotypic adhesion, loss of mesenchymal markers, and reduced in vivo invasive and metastatic potential, rendering them more susceptible to apoptosis [35]. The connection between GPC3 and the Wnt pathway is described in detail in a review article. GPC3 plays a role in the progression of hepatocellular carcinoma by interacting with Wnt signaling proteins and growth factors [36].

GPC3 has been found to inhibit both the canonical Wnt and Akt pathways. GPC3 regulates autocrine and paracrine secretion in breast cancer cells by suppressing the canonical Wnt pathway. This inhibition is essential for the GPC3-mediated regulation of migration and adhesion. Furthermore, GPC3 causes the downregulation of various Wnt pathway-associated molecules, possibly by competing with or sequestering Wnt factors to prevent their binding to frizzled receptors [37]. Notably, the expression of this gene was also downregulated in our polyp specimens.

Signaling molecules in the TGF-β pathway, which frequently synergizes with Wnt, can also influence Wnt signaling. Gremlin 1 (GREM1) is a secreted BMP antagonist that inhibits TGF-β signaling by binding BMPs [38]. *GREM1* expression also exhibited a downregulation trend in our polyp specimens.

A recent study investigated apoptosis-related genes (such as *BCL2* and *BAX*) in polyp development. *BCL2* is a known oncogene. The protein it encodes can inhibit apoptosis, and BAX is known to induce low levels of apoptosis. An analysis of EPs showed an increased BCL2/BAX ratio, representing a potential mechanism promoting EP growth [39]. However, we did not observe changes in the expression of related genes in our RNA-seq results.

Epigenetics may also play a role in polyp development. Approximately 14% of the DEGs in our study were lncRNAs, which could affect polyp development by regulating the expression of other genes. A prior study showed that progesterone could regulate EPs by modulating the expressions of long non-coding RNA H19 and microRNA-152 to influence Wnt and β-catenin signaling [2]. Furthermore, the correlation between cell-free DNA methylation profiles and T-cell differentiation in endometrial polyp patients can help explain the pathogenesis of polyps [40].

The heatmap shows the intrinsic individual variation in differential gene expression within adjacent endometrium. The endometrium is a dynamic environment where gene expression fluctuates in response to hormonal changes throughout the menstrual cycle. We aimed to control the timing of polyp resection surgery in participants during the mid-to-late proliferative phase. However, obtaining clinical specimens poses limitations, making precise control challenging. Physiological status, age, and other factors can influence endometrial gene expression. We selected a neighbor visually normal endometrium from the same patient with endometrial polyps as a comparison control to minimize the effects of interindividual variation. However, an endometrium prone to polyp formation may differ somewhat from a completely normal endometrium, which should be considered.

The development of EPs appears to be influenced by the collective effects of metabolic, pharmacological, and environmental factors. Previous literature has described various contributing elements, including enzymes, diabetes, obesity, hypertension, age, menopausal status, and steroid hormone receptors [41,42]. These factors can affect endometrial gene expression to regulate mechanisms of polyp formation. Our results present some potential clues, though the actual regulatory mechanisms still require further in-depth investigations.

We acknowledge some limitations in our study. First, the number of clinical samples was limited, especially the difficulty in obtaining normal endometrial samples adjacent to polyps, resulting in an insufficient sample amount. Therefore, beyond quantitative RT-PCR, we could hardly perform further validation analyses. Second, due to the heterogeneity and complexity of the clinical samples, we could not standardize the physiological parameters and medication conditions for all subjects. To maximize similarity, we collected specimens from as close a time period as possible and analyzed the differences in gene expression between polyps and adjacent endometrial tissues from the same subject.

This study is the first to examine differences in gene expression between polyp tissues and adjacent endometrial tissues using RNA-seq analysis. By studying the pathogenesis of polyps at the genetic level and their potential effects on embryo implantation during pregnancy, this work can provide an essential reference for clinical treatment guidelines and infertility treatments.

## 4. Materials and Methods

### 4.1. Sample Collection and Processing

Patients undergoing infertility evaluations at the Obstetrics and Gynecology Clinic of Mackay Memorial Hospital in Tamsui were identified with EPs through ultrasound or hysterosalpingography examinations, followed by subsequent polypectomy procedures. The age range of the patients was from 20 to 45 years old. Patients were excluded if they had specific medical conditions and other relevant factors, including previous chemotherapy or radiotherapy, endometrial cancer, use of intrauterine devices, and oral contraceptive use. We enrolled twelve women and obtained samples from EPs and adjacent endometrial tissue. During the polypectomy procedure, the excised polyps and adjacent endometrial tissues were sent for a pathological examination to assess potential carcinogenic risks. Concurrently, a smaller portion of these samples was collected into cryovials filled with liquid nitrogen for subsequent RNA extraction and further analyses after sample collection. The study received approval from the Institutional Review Board of Mackay Memorial Hospital (approval number: 21MMHIS024e), and informed consent was obtained from all participants.

### 4.2. RNA Extraction

Total RNA was extracted using the RNeasy Kit (catalog no. 74104, Qiagen, Hilden, Germany) and then reconstituted in 30 μL of UltraPure DNase/RNase-Free Distilled Water (Invitrogen, Carlsbad, CA, USA). RNA concentration was measured using the Qubit RNA HS Assay Kit (catalog no. Q32855, Thermo Fisher Scientific, Waltham, MA, USA) with the Invitrogen Qubit 4 Fluorometer (Thermo Fisher Scientific). The extracted RNA was then stored at −80 °C.

### 4.3. RNA-seq

Initial RNA samples were adjusted to a 5 ng/μL concentration for a quality assessment. A total of 1–5 μL of each sample was obtained for microfluidic electrophoresis analysis using the Agilent 2100 Bioanalyzer (Agilent Technologies, Beijing, China) with the RNA 6000 nano kit (Agilent Technologies) to evaluate the RNA sample quality. To generate RNA-seq libraries, 1 µg of each sample was used for library construction with the Universal Plus mRNA-Seq with NuQuant (Tecan, Morgan Hill, CA, USA). Library concentration was measured using the Qubit Fluorometer (Invitrogen) with the dsDNA High Sensitivity kit (Invitrogen), and library quality was assessed using the Agilent 2100 Bioanalyzer (Agilent Technologies) with the DNA 1000 kit (Agilent Technologies). Sequencing was then initiated after confirming a satisfactory library quality.

Sequencing was performed on the NovaSeq 6000 system using a 2 × 151 bp paired-end sequencing strategy. The initial sequencing data were quality-controlled, and adapter sequences, low-quality reads, and ambiguous nucleotides were trimmed using the CLC Genomics Workbench v10 (Qiagen). The clean reads from the trimmed dataset were then mapped to the human reference genome (Hg38) using the CLC Genomics Workbench. The mapping parameters were the following: a mismatch cost of 2, an insertion cost of 3, a deletion cost of 3, length fraction of 0.5, and a similarity fraction of 0.8. Gene expression abundance was directly proportional to the number of reads obtained. Fragments per kilobase of transcript per million fragments mapped (FPKM) fragments were calculated to quantify the expression level of individual genes. This metric represents the number of reads mapped to a specific gene per kilobase of exon length per million mapped reads. The RNA-seq data from this study were deposited in the Sequence Read Archive (SRA) database under accession number PRJNA1033797.

### 4.4. Bioinformatics Analysis

We used the prcomp function in R packages to perform a PCA on our RNA-seq dataset. DEGs were identified using R packages DESeq2 (v.1.16.1), which employed a statistical approach based on the negative binomial distribution. Genes with an adjusted *p*-value of less than 0.05 and a fold change of 1.5 or higher were deemed significantly differentially expressed. This analysis was conducted with CLC Genomics Workbench version 10, which included PCA capabilities for the RNA-Seq data.

For the heatmap analysis, a Z-score normalization was applied to RPKM values of genes that showed at least a 1.5-fold change and met the significance threshold with a *p*-value of 0.05 or lower.

We utilized the DAVID online tool to delineate the functional and pathway enrichment analysis of the targeted DEGs [43]. This analysis included both GO and KEGG pathway enrichment assessments. We applied a false discovery rate criterion for significance at *p* < 0.05.

In the PPI network construction and hub gene identification, the STRING database (version 12.0, https://string-db.org/ (accessed on 18 December 2023)) was used to predict the PPI network. We selected PPI pairs with a combined score greater than 0.4. The network visualization was accomplished using Cytoscape software (version 3.10.1), and within this environment, the CytoHubba plugin was employed to compute the connectivity degree of each protein node. The ten proteins with the highest degrees were designated as hub genes.

### 4.5. qRT-PCR

The top ten genes identified from the bioinformatics analysis were further validated using qRT-PCR in matching samples of polyp tissue and adjacent tissue from the same patients. This validation was performed on the LightCycler 96 Real-Time PCR System (Roche). The PCR protocol was set as follows: an initial denaturation at 95 °C for 10 min, followed by 40 amplification cycles at 95 °C for 15 s and 60 °C for 1 min. The Ct (threshold cycle) values were determined automatically by LightCycler 96 software. Relative quantification of RNA expression was calculated using the 2^−ΔΔCt^ method. Primer sequences used are listed in Appendix A. PCR reactions were normalized to the mean expression of three housekeeping genes: YWHAZ, PRDM4, and PUM1. These genes were chosen based on their stable expression as recommended for uterine point-of-care PCR applications in the literature [44,45,46]. They were also selected from our RNA-seq data for their low variability in expression between groups.

### 4.6. Statistical Analysis

For the qRT-PCR analysis, between-group variables were analyzed using independent-sample *t*-tests, as appropriate. Fisher’s exact test was used to compare ratios between the groups. Analyses were performed using GraphPad Prism version 6 (GraphPad, San Diego, CA, USA). A *p*-value < 0.05 wsas considered statistically significant.

## 5. Conclusions

This transcriptomic analysis identified intrinsic gene expression changes in polyps compared with the adjacent endometrial tissues, providing insights into potential mechanisms of pathological overgrowth. Alterations in Wnt signaling and vascular smooth muscle pathways likely contribute to excessive proliferation and impaired vascular/stromal development. The dysfunction of these vital homeostatic pathways can ultimately lead to abnormal bleeding and infertility frequently clinically associated with endometrial polyps. Further investigations into the regulatory relationships between implicated genes and confirmation of additional sample cohorts is warranted. Nonetheless, these findings reveal promising targets involved in polyp formation that can be leveraged for improved diagnostic, preventative, or therapeutic interventions. Elucidating the aberrant molecular signaling underlying this common gynecological condition marks an important step toward the better management and restoration of uterine health.

## Figures and Tables

**Figure 1 ijms-25-02557-f001:**
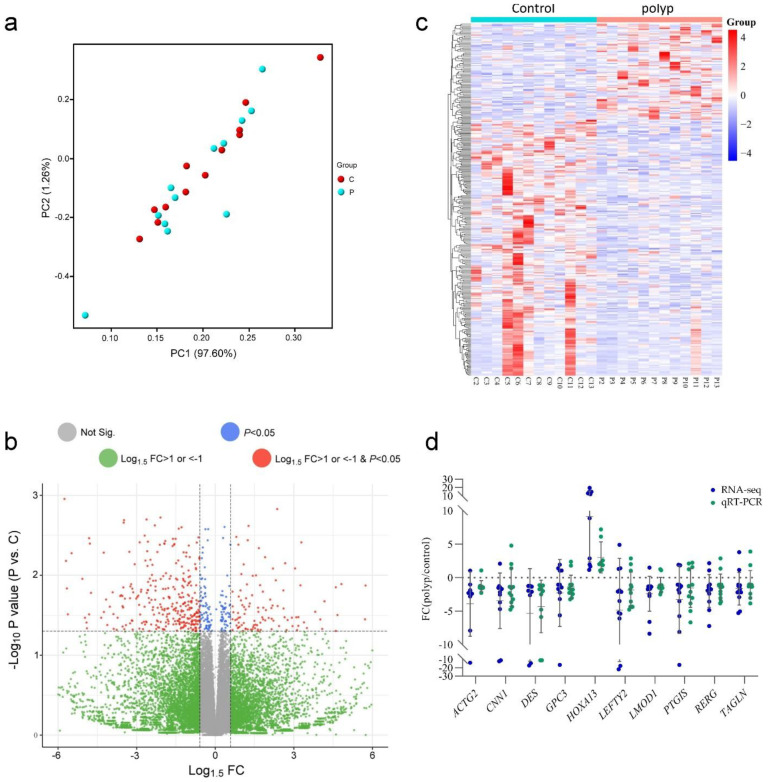
Differences in gene expression between polyps and adjacent endometrial tissues. (**a**) PCA analysis shows gene expression differences between polyp and control groups: red dots represent adjacent endometrial tissues (control, C) and blue dots represent polyp tissues (P). To increase the clarity of the figure, we redrew it based on the original figure. Please see Appendix A for the original figure. (**b**) Volcano plot revealing transcriptional changes (fold change, FC = polyp/control) in polyp tissues and control groups. DEGs were defined as log_1.5_ FC > 1 or < −1 with a *p*-value < 0.05. (**c**) Unsupervised hierarchical clustering heatmap of 322 DEGs. The color depth of red blocks represents the overexpression level of genes. The color depth of blue blocks represents the downregulation level of genes. (**d**) Expression differences (fold changes) of 10 selected genes between polyps and adjacent endometrial tissues were analyzed by qRT-PCR (green dots) and compared with RNA-seq results (blue dots) (*n* = 12).

**Figure 2 ijms-25-02557-f002:**
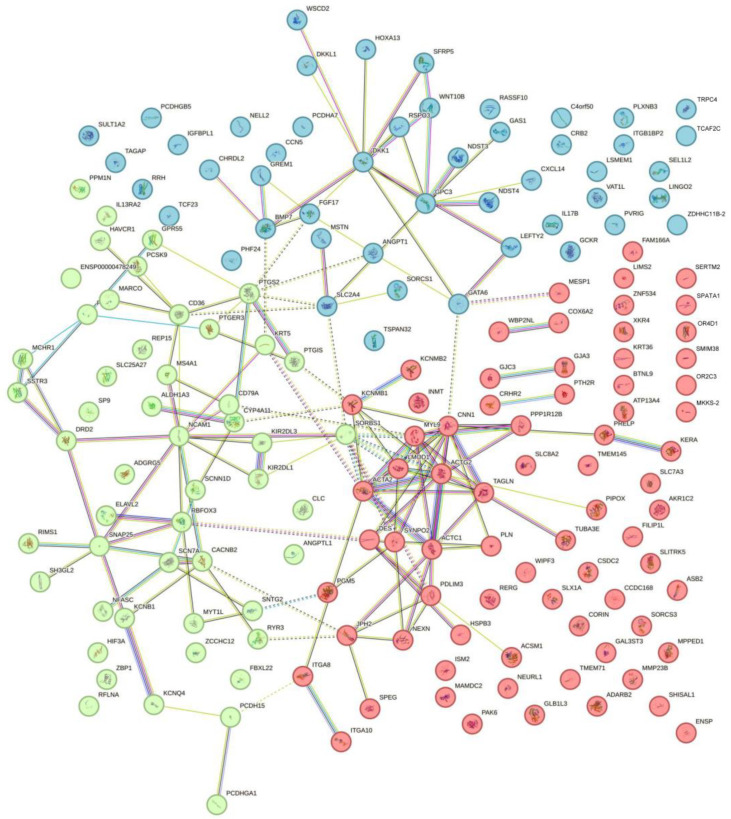
Three PPI subclusters are shown using k-means clustering, represented by three different colors. Subclusters one, two, and three are defined by blue, red, green, with 48, 73, and 51 DEGs, respectively.

**Figure 3 ijms-25-02557-f003:**
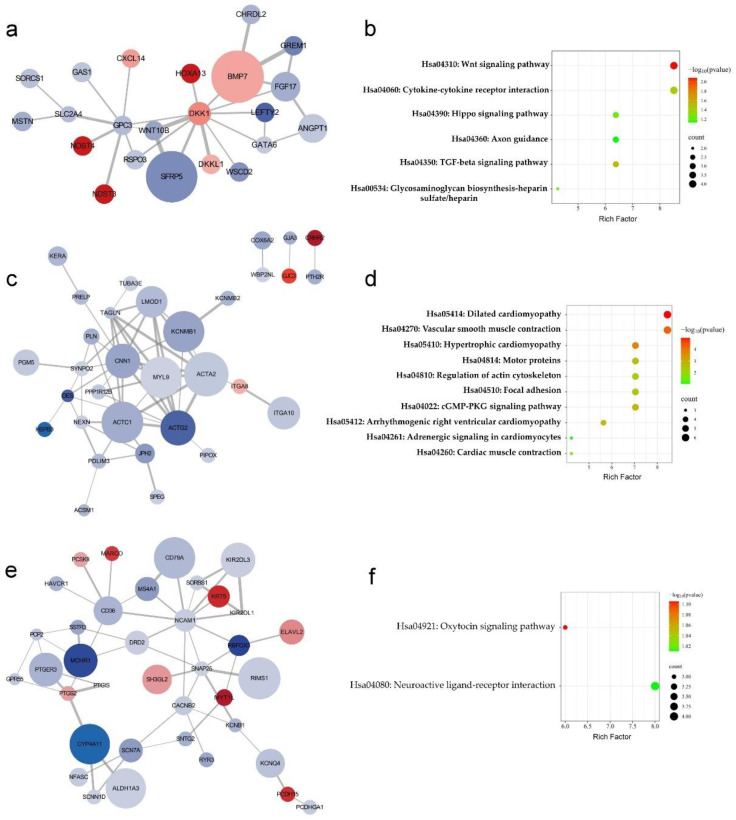
Cytoscape reconstruction and KEGG pathway analysis for the three PPI subclusters. Cytoscape was used to reconstruct individual PPI subclusters one to three ((**a**,**c**,**e**), respectively); KEGG pathway analysis reveals the significant pathways involved in each protein subcluster (**b**,**d**,**f**).

**Figure 4 ijms-25-02557-f004:**
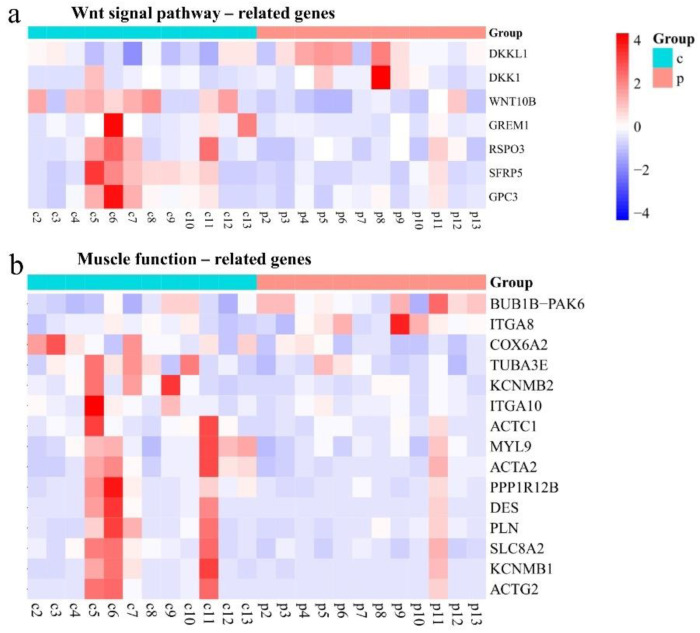
Unsupervised hierarchical clustering heatmap of Wnt signaling pathway (**a**) and muscle function-related genes (**b**).

**Table 1 ijms-25-02557-t001:** GO enrichment analysis for the biological process of the three PPI subclusters.

GOID	Biological Process	*p*-Value	Gene Name
**Subcluster 1**			
GO:060070	canonical Wnt signaling pathway	2.64 × 10^−6^	*WNT10B*, *SFRP5*, *DKKL1*, *GPC3*, *RSPO3*, *DKK1*
GO:0007267	cell–cell signaling	1.33 × 10^−4^	*FGF17*, *GREM1*, *CCN5*, *CXCL14*, *IL17B*, *TSPAN32*
GO:2000096	positive regulation of Wnt signaling pathway, planar cell polarity pathway	1.63 × 10^−4^	*GPC3*, *RSPO3*, *DKK1*
GO:0030210	heparin biosynthetic process	2.48 × 10^−4^	*ANGPT1*, *NDST4*, *NDST3*
GO:0090090	negative regulation of canonical Wnt signaling pathway	4.79 × 10^−4^	*GREM1*, *SFRP5*, *DKKL1*, *GPC3*, *DKK1*
GO:0001707	mesoderm formation	2.31 × 10^−3^	*CRB2*, *BMP7*, *DKK1*
GO:0030326	embryonic limb morphogenesis	2.75 × 10^−3^	*GREM1*, *BMP7*, *DKK1*
GO:0030513	positive regulation of BMP signaling pathway	3.72 × 10^−3^	*CRB2*, *GATA6*, *GPC3*
GO:0010862	positive regulation of pathway-restricted SMAD protein phosphorylation	5.24 × 10^−3^	*MSTN*, *LEFTY2*, *BMP7*
GO:0007165	signal transduction	5.60 × 10^−3^	*FGF17*, *GREM1*, *RASSF10*, *SFRP5*, *TAGAP*, *CCN5*, *ITGB1BP2*, *CXCL14*, *IL17B*
**Subcluster 2**			
GO:0090131	mesenchyme migration	9.30 × 10^−5^	*ACTA2*, *ACTC1*, *ACTG2*
GO:0030239	myofibril assembly	8.31 × 10^−4^	*LMOD1*, *PGM5*, *MYL9*
GO:0007613	memory	2.62 × 10^−3^	*ITGA8*, *PAK6*, *SORCS3*, *SLC8A2*
GO:0007154	cell communication	7.79 × 10^−3^	*GJC3*, *GJA3*, *SLC8A2*
GO:0007612	learning	1.43 × 10^−2^	*PAK6*, *SORCS3*, *SLC8A2*
GO:0030855	epithelial cell differentiation	3.89 × 10^−2^	*TAGLN*, *KRT36*, *AKR1C2*
GO:0005513	detection of calcium ion	4.23 × 10^−2^	*KCNMB1*, *KCNMB2*
GO:0006937	regulation of muscle contraction	4.23 × 10^−2^	*PPP1R12B*, *MYL9*
GO:0007229	integrin-mediated signaling pathway	4.61 × 10^−2^	*ITGA10*, *ITGA8*, *LIMS2*
GO:0006874	cellular calcium ion homeostasis	4.83 × 10^−2^	*PLN*, *ATP13A4*, *SLC8A2*
**Subcluster 3**			
GO:0043269	regulation of ion transmembrane transport	4.60 × 10^−3^	*CACNB2*, *KCNB1*, *KCNQ4*, *SCN7A*
GO:0007166	cell surface receptor signaling pathway	8.07 × 10^−3^	*MARCO*, *ADGRG5*, *CD36*, *MS4A1*, *MCHR1*
GO:0098900	regulation of action potential	9.63 × 10^−3^	*KCNB1*, *CD36*
GO:0031622	positive regulation of fever generation	1.20 × 10^−2^	*PTGER3*, *PTGS2*
GO:0070374	positive regulation of ERK1 and ERK2 cascades	1.80 × 10^−2^	*MARCO*, *GPR55*, *CD36*, *DRD2*
GO:0007626	locomotory behavior	1.87 × 10^−2^	*SNAP25*, *ALDH1A3*, *DRD2*
GO:0060013	righting reflex	2.15 × 10^−2^	*ALDH1A3*, *PCDH15*
GO:0019371	cyclooxygenase pathway	2.15 × 10^−2^	*PTGIS*, *PTGS2*
GO:0007200	phospholipase C-activating G-protein coupled receptor signaling pathway	2.21 × 10^−2^	*PTGER3*, *GPR55*, *DRD2*
GO:0006911	phagocytosis, engulfment	3.20 × 10^−2^	*MARCO*, *CD36*, *HAVCR1*

## Data Availability

The RNA sequencing data used in this study were deposited in the SRA database (https://www.ncbi.nlm.nih.gov/sra/PRJNA1033797, accessed on 19 September 2023) with the accession number: PRJNA1033797 (release date: 1 October 2024).

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
