# Peer review of "Transcriptomic Analysis Reveals Intrinsic Abnormalities in Endometrial Polyps"

_ijms, 2024, doi:10.3390/ijms25052557_

Round 1

Reviewer 1 Report

Comments and Suggestions for Authors

This is an interesting work investigating gene expression differences between endometrial polyp samples and adjacent endometrial tissue by transcriptomic analysis in 12 patients.  The manuscript is clear, the results well exposed and discussed. The study would have been much more powerful if protein expression had also been assessed by proteomic analysis.

Minor points:

Line 98: one gene is missing

Figure 1d is not clear. In the legend, NGS stand for? On lines 108-109 “red dots” and “blue squares” are not present in the figure. Please correct.

Author Response

Q1: This is an interesting work investigating gene expression differences between endometrial polyp samples and adjacent endometrial tissue by transcriptomic analysis in 12 patients. The manuscript is clear, the results well exposed and discussed. The study would have been much more powerful if protein expression had also been assessed by proteomic analysis.

A1: Yes. We agree that proteomic analysis will be more comprehensive for understanding the pathology of endometrial polyps, and we have also planned to analyze the proteome changes in polyps.

Q2: Line 98: one gene is missing.

A2: Yes, the gene LEFTY2 is missing. We have added this gene in line 98. We thank the reviewer for the careful review.

Q3: Figure 1d is not clear. In the legend, NGS stand for? On lines 108-109 “red dots” and “blue squares” are not present in the figure. Please correct.

A3: We have changed "NGS" to "RNA-seq", "red dots" to "green dots", and "blue squares" to "blue dots". Please see the legend in Figure 1d and line 109.

Reviewer 2 Report

Comments and Suggestions for Authors

Dear Authors, 

This manuscript does not supply clear figures, schemes, and tables. This presented style is not good ways for understanding of readers.

I suggest to re-think presentation of figures. This paper should have good quality artwork. The content of the paper is relevant.

Introduction: good content, interesting to the reader.

Discussion: sufficient and log

Methodology: is prepared good.

References: current with good selection

Sincerely

Author Response

Q1: This manuscript does not supply clear figures, schemes, and tables. This presented style is not good ways for understanding of readers. I suggest to re-think presentation of figures. This paper should have good quality artwork. The content of the paper is relevant.

A1: For the convenience of review, the resolution of the PDF file we provide may be insufficient, but it is clear enough from the Word file. We also offer high-resolution TIF files in a single zip archive. An enlarged view of Figure 2 is also included in the Supplementary Material (Figure S2).

Reviewer 3 Report

Comments and Suggestions for Authors

Thank you for inviting me to evaluate the article titled “Transcriptomic Analysis Reveals Intrinsic Abnormalities in Endometrial Polyps”. This study analyzed gene expression differences between EPs and adjacent endometrial tissue to elucidate intrinsic abnormalities promoting pathological overgrowth. The contribution of these observation to the related field is novel. The authors describe their results rationale. I have some minor comments as below.

1. In the bioinformatics analysis of the methods parts, the author needs to provide more information about what R package they used and how they processed these RNA-seq data in the bioinformatic analysis.

2. In the PCA analysis of figure 1a, I suggest the author could only use PC1 and PC2, instead of the 3D plot in the figure, which could help the reader to see the distribution of these samples more clearly.

3. In the heatmap of 4A-B, I suggest the author could use the boxplot to compare these pathways related genes and put them as supplements.

4. The author need to discuss the limitations of this study at the end of discussion part.

5. The language needs to be revised.

Author Response

Q1: In the bioinformatics analysis of the methods parts, the author needs to provide more information about what R package they used and how they processed these RNA-seq data in the bioinformatic analysis.

A1: In the bioinformatics analysis of the Methods section, we describe using the R packages to analyze principal component analysis (PCA) and identify differentially expressed genes (DEGs). Please see lines 364~371 and Lines 377~378.

Q2: In the PCA analysis of figure 1a, I suggest the author could only use PC1 and PC2, instead of the 3D plot in the figure, which could help the reader to see the distribution of these samples more clearly.

A2: To help the reader more clearly see the distribution of gene expression patterns across these samples, we accept the reviewer's suggestion to replace the 3D plot with a 2D plot. Due to the poor resolution of the original image after R analysis, we redrew it based on the original image and placed the original image in supplementary materials (Figure S1).

Q3: In the heatmap of 4A-B, I suggest the author could use the boxplot to compare these pathways related genes and put them as supplements.

A3: We have included Figures 4a and b boxplots to the Supplemental materials (Please see Figures S6).

Q4: The author need to discuss the limitations of this study at the end of discussion part.

A4: We have added a paragraph to discuss the limitations of this study. Please see lines 316~323.

Q5: The language needs to be revised.

A5: As other reviewers have noted, they did not identify any issues with the English. However, to further ensure the quality, we have asked a fluent English-writing colleague to review the manuscript.

Reviewer 4 Report

Comments and Suggestions for Authors

The manuscript performs transcriptomic analysis on endometrial polyps in patients demonstrating changes in gene expression. The authors have two pairs of the endometrial polyps and the surrounding endometrial tissue from infertile women and performed analysis for differentially expressed genes. The authors have used various bioinformatics software for data analysis to present their findings. Furthermore, the authors have validated gene expression in these samples by performing qRT-PCR. Overall, the work is well intentioned, and the results support the conclusions. Rigorous bioinformatics analytics have been used in the study. The weakness noted in the work is that the authors should have validated the results using the published database of utilizing other endometrial polyps and tissue for cross validation.

Author Response

Q1: The manuscript performs transcriptomic analysis on endometrial polyps in patients demonstrating changes in gene expression. The authors have two pairs of the endometrial polyps and the surrounding endometrial tissue from infertile women and performed analysis for differentially expressed genes. The authors have used various bioinformatics software for data analysis to present their findings. Furthermore, the authors have validated gene expression in these samples by performing qRT-PCR. Overall, the work is well intentioned, and the results support the conclusions. Rigorous bioinformatics analytics have been used in the study. The weakness noted in the work is that the authors should have validated the results using the published database of utilizing other endometrial polyps and tissue for cross validation.

Q1: We acknowledge that the weakness of this study warrants further verification of the involvement of the Wnt signaling pathway and vascular smooth muscle regulation in the pathological mechanism of endometrial polyp formation. Since there is no RNA-seq dataset of endometrial polyps available in published databases, except for the one generated in this study, future studies should recruit a larger cohort of endometrial polyp and tissue samples for cross-validation.

Round 2

Reviewer 3 Report

Comments and Suggestions for Authors

Thank you for revising the manuscript. The author had already revised what I'm concerning.

Reviewer 4 Report

Comments and Suggestions for Authors

The comments provided by the authors that there is lack of published dataset on the endometrial polyps is acceptable. The authors have provided limitation of the study in the discussion section.